# Early Cellular and Humoral Responses Developed in Oncohematological Patients after Vaccination with One Dose against COVID-19

**DOI:** 10.3390/jcm11102803

**Published:** 2022-05-16

**Authors:** Sara Rodríguez-Mora, Magdalena Corona, Montserrat Torres, Guiomar Casado-Fernández, Javier García-Pérez, Fernando Ramos-Martín, Lorena Vigón, Mario Manzanares, Elena Mateos, Fernando Martín-Moro, Alejandro Zurdo-Castronuño, María Aranzazu Murciano-Antón, José Alcamí, Mayte Pérez-Olmeda, Javier López-Jiménez, Valentín García-Gutiérrez, Mayte Coiras

**Affiliations:** 1Immunopathology Unit, National Center of Microbiology, Instituto de Salud Carlos III, 28029 Madrid, Spain; srmora@isciii.es (S.R.-M.); m.torres@isciii.es (M.T.); guiomar.casado@externos.isciii.es (G.C.-F.); fernando.ramos@isciii.es (F.R.-M.); lvhernandez@isciii.es (L.V.); mario.manzanares@isciii.es (M.M.); emateo@isciii.es (E.M.); 2Hematology and Hemotherapy Service, Instituto Ramón y Cajal de Investigación Sanitaria (IRYCIS), Hospital Universitario Ramón y Cajal, 28034 Madrid, Spain; madacorona@gmail.com (M.C.); fmartinmoro@usal.es (F.M.-M.); alexzurdo6@gmail.com (A.Z.-C.); jljimenez@salud.madrid.org (J.L.-J.); 3AIDS Immunopathology Unit, National Center of Microbiology, Instituto de Salud Carlos III, 28029 Madrid, Spain; eoaz@isciii.es (J.G.-P.); ppalcami@isciii.es (J.A.); 4Family Medicine, Centro de Salud Doctor Pedro Laín Entralgo, 28924 Alcorcón, Spain; aranzazu.murciano@salud.madrid.org; 5Serology Service, Instituto de Salud Carlos III, 28029 Madrid, Spain; mayteperez@isciii.es

**Keywords:** COVID-19 vaccine, hematological malignancies, cytotoxic response, humoral response, early immune

## Abstract

Individuals with oncohematological diseases (OHD) may develop an impaired immune response against vaccines due to the characteristics of the disease or to its treatment. Humoral response against SARS-CoV-2 has been described to be suboptimal in these patients, but the quality and efficiency of the cellular immune response has not been yet completely characterized. In this study, we analyzed the early humoral and cellular immune responses in individuals with different OHD after receiving one dose of an authorized vaccine against SARS-CoV-2. Humoral response, determined by antibodies titers and neutralizing capacity, was overall impaired in individuals with OHD, except for the cohort of chronic myeloid leukemia (CML), which showed higher levels of specific IgGs than healthy donors. Conversely, the specific direct cytotoxic cellular immunity response (DCC) against SARS-CoV-2, appeared to be enhanced, especially in individuals with CML and chronic lymphocytic leukemia (CLL). This increased cellular immune response, developed earlier than in healthy donors, showed a modest cytotoxic activity that was compensated by significantly increased numbers, likely due to the disease or its treatment. The analysis of the immune response through subsequent vaccine doses will help establish the real efficacy of COVID-19 vaccines in individuals with OHD.

## 1. Introduction

Oncohematological patients are at high risk for developing severe COVID-19 complications, with an estimated mortality rate exceeding 30% [1,2,3]. However, not all OHDs had the same level of susceptibility to SARS-CoV-2 natural infection and significant differences have been described in the outcomes of the infection among patients depending on the disease and type of treatment. Individuals with CML appear to have a significantly low mortality rate in general (5.5–13%) [4,5], although differences between patients on treatment with Tyrosine Kinase Inhibitors (TKI) and those on treatment-free remission (TFR) have not been yet characterized. Moreover, in a multicenter study that included 217 CML patients who had been infected with SARS-CoV-2, being 11% of them on TFR, only 47 patients (21.4%) required hospital admission and all of them were on treatment with TKIs [6]. Other OHDs may present significantly higher rates of hospitalization during COVID-19, as is the case of patients with CLL. Although considering that asymptomatic patients are probably underrated, hospital admissions of CLL patients due to severe COVID-19 are much higher than those with CML, with hospitalization rates nearly at 90% and mortality between 31–35% [1,7,8]. There are differences among patients with CLL who are receiving treatments such as Bruton Tyrosine Kinase (BTK) inhibitors and those who are in “watch and wait”, but all are at high-risk of fatal outcomes [8,9]. Around 48% of patients with Multiple Myeloma (MM) infected with SARS-CoV-2 required hospitalization and mortality in these individuals has been established in 24–33% [10,11]. Although no differences have been reported between patients with MM who had received a prior Autologous Stem Cell Transplant (ASCT) or other different therapies such as proteasome inhibitors, immunomodulatory agents, and anti-CD38 antibodies, there are significant worse outcomes in patients with active or progressive disease comparing to partial or complete responses. Finally, patients with Allogeneic Stem Cell Transplant (Allo-HSCT) are also high-risk patients for developing severe SARS-CoV-2 illness, with 28.4% admissions to the ICU and 22.5% of mortality [12]. Among recipients of Allo-HSCT, infection with SARS-CoV-2 within the first year after transplantation and ongoing immunosuppressive treatment have been described as predictive factors for admission to the ICU, but only time since transplant remains as a predictor for mortality.

For this reason, vaccination is recommended, even though the efficacy is expected to be impaired as happens with other vaccines studied in different oncohematological diseases [6,13,14,15,16,17,18,19]. Many real-life studies have been published lately describing a suboptimal humoral response in immunocompromised patients against SARS-CoV-2 vaccines. As expected, B-cell malignancies present low seroconversion, with rates that vary between 39–55% along with decreased titers and neutralizing capacity of the antibodies [20,21,22,23,24], whereas seroconversion in hematopoietic stem cell transplant recipients has been described to be between 78 and 85% [25]. However, most of these studies have focused exclusively on the humoral response, with scarce evidence integrating both cellular and humoral responses, which is known to be essential to control COVID-19 infection and avoid the progression to a more severe disease [26,27]. Recently, a few studies have provided data on the cellular response elicited by immunocompromised patients, with rates varying between 45 and 80% [28,29,30], which demonstrates that the cellular response does not always correlate with seroconversion as patients may present T-cell responses in the absence of antibodies against SARS-CoV-2 in plasma. However, these results only derive from the release of pro-inflammatory cytokines such as IFN©, TNFα or IL-2, after stimulation with COVID-19 antigens, without providing complete information about the capacity to generate an effective direct cytotoxic response or the antibody-dependent cytotoxicity against SARS-CoV-2-infected cells.

In this study, we analyzed both humoral and cellular early immune responses elicited by individuals with different OHD after receiving one dose of a vaccine against SARS-CoV-2 infection. This information may be essential to understand how the immune response is developed after COVID-19 vaccination in these patients and also to define the booster vaccine schedule that would be more beneficial for each OHD.

## 2. Materials and Methods

### 2.1. Study Populations

Fifty-two patients with different OHD were recruited in the Hematology and Hemotherapy Service at the University Hospital Ramon y Cajal between February and June 2021. All patients were older than 18 years old, they did not have previous COVID-19 infection and were vaccinated through the Spanish Vaccination Program with one dose of a vaccine approved at the time for these patients: COMIRNATY (BNT162b2, BioNTech-Pfizer), Spikevax (mRNA-1273, Moderna), or Vaxzevria (AZD1222, AstraZeneca). Fourteen healthy-donor vaccinated with COMIRNATY (BioNTech-Pfizer) and COVID-19 Vaccine Janssen (Ad26.COV2-S, Janssen) were also included.

Blood samples were obtained prior to vaccination and 3 weeks after receiving the first dose of the vaccine. The absence of infection with SARS-CoV-2 previous to vaccination was determined by basal serology against COVID-19 in the first blood sample in order to discard asymptomatic infection.

### 2.2. Samples Processing and Materials

Blood samples were immediately processed by centrifugation through Ficoll-Hypaque gradient (Pharmacia Corporation, North Peapack, NJ, USA) and peripheral blood lymphocytes (PBMCs) and plasmas were isolated and cryopreserved until the moment of analysis. Raji cell line (ATCC CCL-86) was provided by the existing collection of the Instituto de Salud Carlos III (Madrid, Spain). Vero E6 (African green monkey kidney) cell line (ECACC 85020206) was kindly provided by Dr. Antonio Alcami (CBM Severo Ochoa, Madrid, Spain). Vero E6 and HEK-293T (National Institute for Biological Standards and Control [NIBSC]) cells were cultured in DMEM supplemented with 10% FCS, 2 mM L-glutamine, and 100 units/mL penicillin/streptomycin (Lonza, Basel, Switzerland).

### 2.3. Phenotyping of B Lymphocytes

Subpopulations of B cells (CD3^−^CD19^+^) were analyzed by flow cytometry after staining of surface markers CD10, CD27, CD20, and CD21 as follows: immature or transitional cells (CD10^+^CD27^−^); naïve B cells (CD10^−^CD27^−^CD21^high^); tissue-like memory cells (CD10^−^CD27^−^CD21^low^); resting memory cells (CD10^−^CD27^+^CD21^high^); activated memory cells (CD10^−^CD27^+^CD21^low^); and plasmablasts (CD27^++^CD20^−^CD21^low^) [31]. Antibodies CD3-PE, CD10-BV421, CD19-BV711, CD20-AlexaFluor700, CD21-FITC, and CD27-PercP-Cy5.5 were purchased from BD Biosciences (San Jose, CA). Data acquisition was performed in a BD LSRFortessa X-20 flow cytometer and FACS Diva software v6.0 (BD Biosciences, San Jose, CA, USA). FlowJo software v10.8 (Tree Star Inc., Ashland, OR, USA) was used for data analysis.

### 2.4. SARS-CoV-2 Serology

IgG antibodies against the S protein of SARS-CoV-2 were analyzed in plasma of all individuals by Euroimmun Anti-SARS-CoV-2 ELISA Assay (Euroimmun, Lübeck, Germany). Semi-quantitative results were analyzed by calculating the ratio of extinction of each plasma sample over the calibrator. Results were considered positive with IgG titer >1.1; values between 0.8 and 1.1 were considered undetermined; and values <0.8 were considered negative. Borderline data were considered positive.

### 2.5. Pseudotyped SARS-CoV-2 Neutralization Assays

One single-cycle, pseudotyped SARS-CoV-2 virus (pNL4-3Δenv_SARS-CoV-2-SΔ19(G614)_Ren) was synthesized by co-transfection of HEK-293T cells with vector pNL4-3Δenv_Ren that expresses HIV-1 genome without env gene and Renilla luciferase gene as reporter [32], together with vector pcDNA3.1-SARS-CoV-2-SΔ19 that expresses SARS-CoV-2 S glycoprotein without the last 19 amino acids (QHU36824.1) [33]. Co-transfection with vector pcDNA-VSV-G, which expressed spike (S) glycoproteins of vesicular stomatitis virus (VSV), was used as control of specificity. The concentration of HIV-1 p24/Gag antigen in cell culture supernatants was quantified 48 h post-transfection by Elecsys HIV AG (Roche Diagnostic, Basel, Switzerland).

Plasma neutralization activity was measured by pre-incubation for 1 h at 37 °C of pNL4-3Δenv_SARS-CoV-2-SΔ19(G614)_Ren pseudovirus (10 ng p24 Gag per well) with 4-fold serial dilutions (1/32 to 1/8192) of decomplemented IgG-positive plasma from the recruited patients with oncohematological disorders and healthy donors as previously described [34]. This mixture was then added to a monolayer of Vero E6 cells and incubated for 48 h. Vero E6 cells were then lysed, and viral infectivity was assessed by measuring Renilla luciferase activity (Renilla Luciferase Assay, Promega, Madison, WI, USA) with 96-well plate luminometer Centro XS3 LB 960 with MikroWin 2010 software (Berthold Technologies, Baden-Württemberg, Germany). The titers of neutralizing antibodies were represented as 50% inhibitory dose (ID50), which is the highest dilution of plasma that resulted in a 50% reduction of luciferase activity compared to control without serum, using non-linear regression in GraphPad Prism Software v9.3 (GraphPad, Inc., San Diego, CA, USA).

### 2.6. Antibody-Dependent Cellular Cytotoxicity Assay

Raji cell line was used as target to measure antibody-dependent cellular cytotoxicity (ADCC) capacity of PBMCs of patients and donors, as described before [35]. Briefly, Raji cells were previously labeled with PKH67 Green Fluorescent Cell Linker (Merck KGaA, Darmstadt, Germany) and then coated with rituximab (50 μg/mL) (Selleckhem, Houston, TX, USA) for 4 h. Labeled Raji cells were then co-cultured for 18 h with PBMCs (1:2 ratio) from the recruited patients and healthy donors. Apoptosis of Raji cells was determined by staining with Annexin V conjugated with phycoerythrin (PE) (Immunostep, Salamanca, Spain). Data acquisition was performed in a BD LSRFortessa X-20 flow cytometer and FACS Diva software (BD Biosciences). FlowJo software (Tree Star Inc.) was used for data analysis.

### 2.7. Direct Cellular Cytotoxicity Assay

Due to D614 SARS-CoV-2 viruses were the majority of the earliest variants detected in Spain within clade 19B [36], a mutant clone with D614G change was created by site-directed mutagenesis in pNL4-3Δenv_SARS-CoV-2-SΔ19(G614)_Ren pseudovirus [37]. For analysis of direct cellular cytotoxicity (DCC), Vero E6 cells were infected with equal amounts of both one cycle pseudotyped viruses D614 and G614 (100 ng p24 Gag/well) and then plated onto 48-well plates. After 48 h of incubation, Vero cells were co-cultured for 1 h with PBMCs from patients and healthy donors (ratio 1:10). After detaching Vero monolayer with trypsin-EDTA solution (Sigma Aldrich-Merck, Darmstadt, Germany), caspase-3 activity was measured by luminescence using Caspase-Glo 3/7 Assay system (Promega). Cytotoxic cell populations such as Natural Killer (NK), NKT and TCRγδ+ cells were analyzed in the supernatants using specific conjugated antibodies: CD3-PE, CD56-BV605, CD16-PercP, CD8-APC H7, and TCRγδ-FITC (BD Biosciences). Data acquisition was performed in a BD LSRFortessa X-20 flow cytometer and FACS Diva software (BD Biosciences). FlowJo software (Tree Star Inc.) was used for data analysis.

### 2.8. Statistical Analysis

Statistical analysis was performed using GraphPad Prism 8.0 (GraphPad Software Inc., San Diego, CA, USA). Quantitative variables were represented as mean and standard deviation of the mean (SEM). Significance in the comparison between groups was analyzed using one-way ANOVA and Tukey’s multiple comparisons test. The unpaired, nonparametric Mann–Whitney test was applied to compare between pre- and post-vaccination samples. *p* values (*p*) < 0.05 were considered statistically significant in all comparisons.

## 3. Results

### 3.1. Patients’ Cohorts

A total of 52 patients were included in this study and segregated in four groups depending on diagnosis: MM (n = 15), CML (n = 10), CLL (n = 15), and Allo-HSCT (n = 12). Baseline demographic and clinical characteristics of these individuals are summarized in Table 1.

In the cohort of CLL, 8 individuals (53.4%) were receiving active treatment: 4 of them with Bruton Tyrosine Kinase (BTK) inhibitors, 3 with Rituximab and Venetoclax and 1 with Rituximab and Idelalisib; whereas 7 individuals (46.6%) were on a Watch and Wait follow-up. A total of 11 individuals (73.3%) were vaccinated with a mRNA-based vaccine and 4 individuals (26.7%) were vaccinated with Vaxzevria (Astra Zeneca).

In the cohort of CML, 40% of the patients had discontinued treatment with TKIs after achieving prolonged deep molecular response. Seven individuals (70%) were vaccinated with a mRNA-based vaccine and three individuals (30%) received Vaxzevria.

In the cohort of MM, 6 (40%) individuals were on maintenance treatment after ASCT, with a median time since transplant of 32 months (IQR 23–60). The rest of the MM patients were non-ASCT candidates under a first-line treatment: three individuals with lenalidomide and dexamethasone (Ld), four under daratumumab, bortezomib, melphalan and prednisone (DVMP), one with cyclophosphamide, bortezomib and dexamethasone (CyBorD) and one individual with daratumumab, lenalidomide, carfilzomib and dexamethasone (KDRd). All individuals were vaccinated with a mRNA-based vaccine.

In the cohort of Allo-HSCT patients, 6 individuals (50%) were receiving immunosuppressive treatment for active graft versus host disease (GvHD). The median time since allogeneic transplant was 36 months (IQR 21–44). All individuals were vaccinated with a mRNA-based vaccine.

A total of 14 healthy donors were also recruited for this study, with a median age of 52 years (IQR 46.25–58.25). Fifty percent of the donors were male, and 6 (42.8%) individuals were vaccinated with COMIRNATY (BioNTech-Pfizer), whereas 8 (57.1%) individuals received COVID-19 Vaccine Janssen (Ad26.COV2-S, Janssen).

### 3.2. Early Serological Response after One Dose of SARS-CoV-2 Vaccine

A single-dose of COVID-19 vaccine elicited IgG seroconversion in 42.8% of healthy donors versus 13.33% in CLL, 90% in CML, 40% in MM, and 41.6% in Allo-HSCT (Figure 1A). Therefore, all cohorts except CLL presented IgG SARS-CoV-2 titers over the threshold of detection three weeks after the administration of one dose of the vaccine, with statistical significance in the difference between pre- and post-vaccination samples in all OHD cohorts. Interestingly, whereas individuals with MM and Allo-HSCT showed similar levels of IgGs than healthy donors, patients with CML achieved a very potent humoral response, with early IgGs titers against COVID-19 that were increased 5.4- (*p* < 0.0001), 2.9- (*p* = 0.0026), and 2.8- (*p* = 0.0058) fold in comparison with individuals with CLL, MM and HSCT, respectively, and also 3.3-fold higher than in healthy donors (*p* = 0.0012).

The neutralizing capacity against SARS-CoV-2 of IgGs from the individuals with detectable antibodies in plasma was also analyzed. All healthy donors (7/7; 100%) with detectable IgGs developed neutralizing antibodies above the threshold of detection, whereas only 56% of the individuals with OHD showed neutralizing IgGs: 1/3 (33%) CLL, 5/9 (55%) CML, 4/7 (57%) MM, and 4/6 (67%) Allo-HSCT (Figure 1B). This neutralizing activity was similar or lower than healthy donors, except for individuals with CML and Allo-HSCT who presented mean levels of neutralizing antibodies that were increased in comparison with healthy donors, although these results did not achieve statistical significance. Only three individuals with CLL developed IgGs against SARS-CoV-2 after one dose of vaccine, and only one developed highly neutralizing antibodies, whereas the other two individuals showed IgGs with neutralizing activity below the threshold of detection.

### 3.3. Early Changes in B Cell Subpopulations after One Dose of SARS-CoV-2 Vaccine

Total levels of B cells did not significantly change within groups after receiving the first dose of the vaccine (Figure 2A). In the comparison between groups, the CLL cohort presented the highest levels of B cells (CD19+), which were increased 5.1- (*p* < 0.0001), 4.2- (*p* < 0.0001), and 3.7-fold (*p* < 0.0001), respectively, in comparison with individuals with CML, MM, and Allo-HSCT.

The most significant differences between groups of individuals with OHD were observed in the subpopulation of naïve B cells (CD10^−^CD27^−^CD21^high^) (Figure 2B). The individuals with CLL and MM showed naïve B cell levels significantly reduced 1.8- (*p* < 0.0001) and 1.3-fold (*p* = 0.0162), in comparison with healthy donors. Moreover, resting memory B cells (CD10^−^CD27^+^CD21^high^) were reduced 1.9- (*p* = 0.0440) and 7.6-fold (*p* = 0.0214) in individuals with CLL and Allo-HSCT, in comparison with healthy donors.

### 3.4. Impaired Antibody-Dependent Cellular Cytotoxicity in PBMCs from Individuals with OHD

ADCC response increased in healthy donors after vaccination, but it overall decreased in patients with OHD, especially in those with Allo-HSCT, which was reduced 2.2-fold (*p* = 0.0252) in comparison with healthy donors (Figure 3).

### 3.5. Viral Neutralization and Direct Cellular Cytotoxicity against Cells Infected with Pseudotyped SARS-CoV-2

The PBMCs of individuals with CLL showed the highest DCC activity against SARS-CoV-2-infected cells, which was even present before vaccination (Figure 4A). This activity was significantly increased in comparison with PBMCs of individuals with Allo-HSCT before (2.3-fold; *p* = 0.0166) and after (2.6-fold; *p* = 0.0020) one-dose vaccination. These results correlated with the measurement of caspase-3 activity in the monolayer of Vero E6 cells, although they did not achieve statistical significance in the comparison between groups (Figure 4B).

### 3.6. Characterization of Cellular Cytotoxic Populations

PBMCs populations that were responsible for DCC response were analyzed and we observed that individuals with CLL showed lymphopenia, with the levels of CD3^+^ T cells reduced 3.4-fold (*p* < 0.0001) before vaccination, in comparison with healthy donors. This CD3^+^ lymphopenia was maintained after vaccination (*p* < 0.0001) (Figure 5A). Individuals with Allo-HSCT also showed levels of CD3^+^ that were reduced 1.7-fold (*p* = 0.0251), whereas individuals with CML and MM showed similar levels to healthy donors. Despite CD3^+^ lymphopenia, individuals with CLL and Allo-HSCT showed levels of CD8^+^ T cells that were increased about 2.0-fold in both cases, before (*p* = 0.0038 and *p* = 0.0023, respectively) and after (*p* = 0.0018 and *p* = 0.0238, respectively) vaccination (Figure 5B, left graph). Activation of CD8^+^ T lymphocytes was evaluated through the expression of the degranulation marker CD107a, which was significantly reduced 1.5- (*p* = 0.0271) and 1.8-fold (*p* = 0.0014) in individuals with CLL and MM, respectively, in comparison with healthy donors, and did not change after vaccination (Figure 5B, right graph).

There was a great variability between the levels of NK cells (CD3^-^CD56^+^) between the cohorts and pre- and post-vaccination samples, being the individuals with CLL those with more reduced levels before vaccination (Figure 5C, left graph). However, they showed similar levels of CD107a expression as healthy donors before vaccination, although they decreased after vaccination (Figure 5C, right graph). After vaccination, the expression of CD107a in NK cells in individuals with MM and Allo-HSCT was reduced 4.3- (*p* = 0.0028) and 3.6-fold (0.0137), in comparison with healthy donors. The levels of NKT cells did not significantly change between the cohorts of individuals with OHD and healthy donors (data not shown).

Individuals from all cohorts had similar levels of CD3^+^CD8^+^TCRγδ^+^ cells (Figure 6A, left graph), but the expression of CD107a was mostly reduced in these cells of individuals with MM (3.0-fold; *p* = 0.0158) before vaccination, in comparison with healthy donors (Figure 6A, right graph), although it was significantly increased after one-dose vaccination (2.1-fold; *p* = 0.0374). Moreover, individuals with MM showed significantly increased levels of CD3^+^CD8^-^TCRγδ+cells before vaccination (4.4-fold; *p* = 0.0196) (Figure 6B, left graph), in comparison with healthy donors, and these cells also showed high expression of CD107a in some individuals (Figure 6B, left graph), although these levels returned to similar levels as healthy donors after vaccination. Individuals with CML showed a steep reduction in the expression of CD107a CD3^+^CD8^+^TCRγδ^+^ cells after vaccination (2.2-fold; *p* = 0.0368) (Figure 6A, right graph) and nearly undetectable levels in CD3^+^CD8^-^TCRγδ^+^ cells (Figure 6B, left graph).

## 4. Discussion

Individuals with OHD are known to present impaired immune responses to several infectious agents and their vaccines, such as influenza, varicella Zoster and hepatitis B [6,13,14,15,17,18,19,38,39,40]. Accordingly, diverse types of OHD, as well as the type of treatment that these patients receive, may influence the immune response against these vaccines, including SARS-CoV-2 vaccines. Therefore, the efficacy of SARS-CoV-2 vaccines in these individuals has been a major concern since the beginning of the massive vaccination. Many studies that evaluate the quality of the immune response against COVID-19 vaccines in individuals with OHD have been published lately. However, these studies mostly describe the humoral response, with low characterization of the cellular immune responses, which are usually limited to the evaluation of T-cell reactivity through cytokines quantification after antigen stimulation [28,29,30], whereas the information regarding functional cellular cytotoxicity is limited to small series of patients for a single disease [41].This T-cell reactivity achieved 86% rate in individuals with myeloproliferative neoplasms (MPN) after a single dose [28] and 87.3% and 83% in MM and CLL, respectively [30] after two doses, whereas the response was widely estimated to be 19–73% in patients with Allo-HSCT [30,42]. These results indicate that the cellular immune response in individuals with OHD might be underestimated due to the poor seroconversion obtained after vaccination. Therefore, in this study, we analyzed not only the early humoral response developed by four groups of individuals with OHD after receiving one dose of an authorized COVID-19 vaccine, but also the potency and quality of the cellular cytotoxic immune response in comparison with healthy donors.

The seroconversion rates in our cohorts were variable depending on each OHD, being under the limit of detection in individuals with CLL but excellent in individuals with CML, who showed significantly higher IgG titers than healthy donors. Other studies corroborate this low seroconversion rates for CLL patients, estimated at 18% following a single dose [43] and that is most likely influenced by the severe B cell impairment in CLL patients treated with B cell depleting or targeted therapies such as anti-CD20 or BTK inhibitors [39,44]. Therefore, although individuals with CLL showed the highest levels of B cells due to the disease, significant alteration of B cell subpopulations was found, such as the reduction of naïve and resting memory B cells, which would entail an altered B cell activity and the reduced levels of seroconversion. Conversely, the excellent early humoral response (90% of seroconversion rate) developed by individuals with CML after one-dose vaccination against SARS-CoV-2, despite the B cell lymphopenia, was in accordance with previous studies that described good seroconversion rates among patients with chronic MPN and CML [45], being 54–71% after one single dose of vaccine [43]. This difference in the humoral responses between CLL and CML may also be related to the potent immunomodulatory effect induced in the latter during treatment with TKIs [19], which may be conserved even after several years of discontinuing treatment [46] and would explain the low hospitalization rate described for individuals with CML on TFR due to COVID-19 [6]. Moreover, IgGs from individuals with CML also showed overall high neutralization rate, whereas only one individual from our cohort of CLL who was on “watch and wait” developed IgGs against SARS-CoV-2 with a very high neutralizing capacity. In fact, the neutralizing capacity of IgGs is usually reduced in B-cell malignancies, such as MM or Waldestrom disease, where clinically relevant antibodies titers are achieved only in 2–8% of the patients after one single-dose vaccine against SARS-CoV-2 [20,21,23].

The role of IgGs to protect against SARS-CoV-2 infection does not rely only on their neutralizing capacity to protect cells from infection but also in their ability to activate the complement system and ADCC response exerted by NK and CD8+ T cells in order to eliminate the infected cells. In our cohorts, early ADCC activity elicited after a single dose of COVID-19 vaccine was generally decreased in PBMCs from all individuals with OHD, in correlation with the low seroconversion, except for the cohort of individuals with CML who showed similar ADCC response than healthy donors. We also evaluated the DCC activity against SARS-CoV-2 infected cells and observed that all individuals, except those with CML, showed higher DCC response than healthy donors, which effectively eliminated the infected cells. This DCC response was mostly efficient in individuals with CLL, which indicated that the impaired humoral response was counteracted by an effective cellular response in these patients, despite the reduced levels of overall CD3^+^ cells. We did not observe significant changes in the levels of CD8^+^ T cells before and after vaccination in any cohort, but total levels were significantly increased in all individuals in comparison with healthy donors, except for individuals with CML. However, CD8^+^ T cells showed lower expression of the degranulation marker CD107a in all cohorts, which may indicate that the high number of cells could compensate the reduced cytotoxic activity, thereby explaining the higher DCC observed in the PBMCs from individuals with CLL. The low levels of CD8^+^ T cells may imply a reduced memory response that would need subsequent boosters of COVID-19 vaccine in order to generate a sustained protective immune response. On the other hand, the levels of highly cytotoxic CD3^+^CD8^±^TCRγδ^+^ cells were also increased in all cohorts, in comparison with healthy donors. In fact, this subpopulation is known to be increased in oncohematological patients [47,48,49]. The level of more immature CD3^+^CD8^-^TCRγδ^+^ cells was increased after vaccination in individuals from CLL and CML cohorts, whereas in individuals with MM the subpopulation of CD3^+^CD8^+^TCRγδ^+^ cells was not only increased in comparison with healthy donors, but it also showed a significant enhancement in the expression of degranulation markers after one dose of vaccine. On the other hand, NK cells were greatly reduced in individuals with CLL and showed a general decrease in the degranulation capacity, mostly in the cohort of individuals with Allo-HSCT, whereas no changes were observed in NKT cells, in comparison with healthy donors. Consequently, the early cellular immune response developed by individuals with OHD after one dose of vaccine seemed to rely mostly on CD8^+^ T cells. Although these cells showed overall reduced count, their antiviral activity was reinforced by increased levels of functional TCRγδ+ cells.

Therefore, as described in other studies [29,30,45], we demonstrated that seroconversion does not always correlate with cellular immune response after vaccination. Individuals with different OHD showed variable early immune responses after receiving one dose of vaccine against SARS-CoV-2. Unlike healthy donors, who showed an efficient early humoral response but more retarded cellular response, the humoral response was overall impaired in all OHD, except for CML, although it was counteracted by a potent early cellular response, especially in individuals with CLL. This enhanced cellular immune response relied mostly on reduced levels of functional CD8+ T cells but also in unconventional CD8±TCRγδ+ T cells with quite modest cytotoxic activity that may be nonetheless compensated by significantly increased numbers in comparison with healthy donors, most likely as a consequence of the disease. Moreover, an increased DCC antiviral activity was detected before vaccination that was surely contributing to the efficient early cellular response after one-dose vaccination and that may be a result of the high basal levels of cytotoxic populations, likely elicited by the presence of cancerous cells or by the anticancer treatment.

The clinical translation of these results in terms of breakthrough COVID-19 infections in patients with different OHD in response to vaccination remains to be described as these individuals were excluded from the clinical trials [50,51,52]. However, immunocompromised patients show higher risk to develop severe COVID-19 than healthy donors and some therapies, such as anti-CD20 antibodies, could be related to worse outcomes [53]. However, the efficacy of the immune response may also be achieved through the development of cellular immune responses that need to be characterized to fully evaluate the efficiency of COVID-19 vaccination. Accordingly, although vaccine efficacy is very variable and dependent on the OHD, vaccination seems to be useful also in these patients to prevent severe infection, at least partially [54]. It also has been shown that vaccines protection may be decreased by different lineages of the virus. Neutralizing antibody responses are negatively impacted by emerging variants as is the case for the Omnicron or Delta variants [55]. However, booster doses with mRNA vaccines have shown to be effective at inducing high neutralizing titers in immunocompetent patients [56] and seem to be able to avoid severe illness even though breakthrough infections may be common. Moreover, mortality in OHD individuals appears to be decreasing through the consecutive waves [57], which may be related to increasing numbers of patients with partial or complete vaccination schedules.

## 5. Conclusions

Early immune responses after one dose of COVID-19 vaccine may have clinical relevance and confer at least partial protection against SARS-CoV-2 infection or severe COVID-19 in individuals with OHD, likely based on a cellular immunity developed in response to the presence of cancerous cells or to the treatment for the disease in the case of individuals with CLL, MM, or Allo-HSCT, but also based on a potent humoral response in the case of patients with CML. The analysis of the immune responses elicited after subsequent vaccine boosters will help establish the real efficacy of COVID-19 vaccines in individuals with OHD, especially concerning the cellular cytotoxic response.

## Figures and Tables

**Figure 1 jcm-11-02803-f001:**
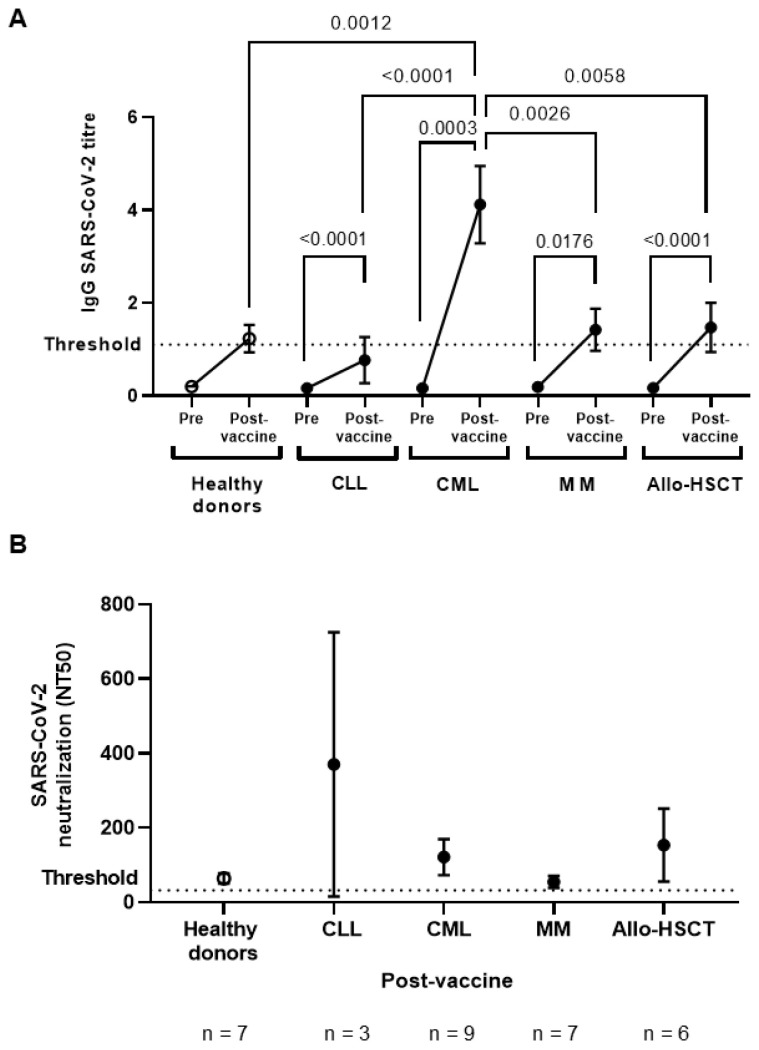
Serological response against SARS-CoV-2 vaccine in plasma from individuals with OHD. (**A**) IgGs titers in plasma from individuals with CLL, CML, MM, or Allo-HSCT chronic lymphocytic leukemia (CLL), chronic myeloid leukemia (CML), multiple myeloma (MM), or allogeneic stem cell transplant (Allo-HSCT) after receiving one-dose of SARS-CoV-2 vaccine, in comparison with healthy donors. (**B**) Neutralizing activity against SARS-CoV-2 of IgGs from individuals with OHD and healthy donors after one-dose vaccination. Each dot in the graph corresponds to mean ± standard error of the mean (SEM). Statistical significance between groups was calculated using one-way ANOVA test and statistical significance within groups was calculated using Mann–Whitney test.

**Figure 2 jcm-11-02803-f002:**
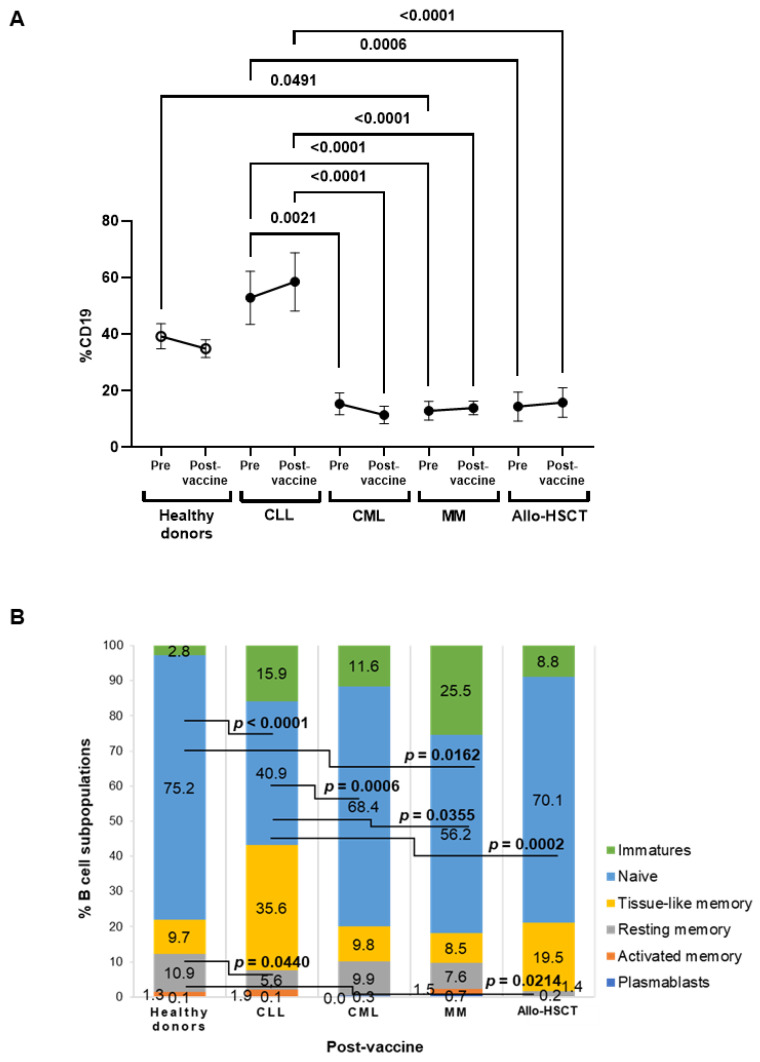
Total levels and subpopulations of B cells in PBMCs of individuals with OHD before and after receiving one-dose vaccine against SARS-CoV-2. (**A**) Total B cell levels in peripheral blood mononuclear cells (PBMCs) of individuals with chronic lymphocytic leukemia (CLL), chronic myeloid leukemia (CML), multiple myeloma (MM), or allogeneic stem cell transplant (Allo-HSCT) before and after receiving the first dose of SARS-CoV-2 vaccine, in comparison with healthy donors. Each dot in the graphs corresponds to mean ± SEM. (**B**) Analysis of B cells subpopulations after receiving one dose of SARS-CoV-2 vaccine in these cohorts. Mean data are represented in bar graphs. Statistical significance between groups was calculated using one-way ANOVA test.

**Figure 3 jcm-11-02803-f003:**
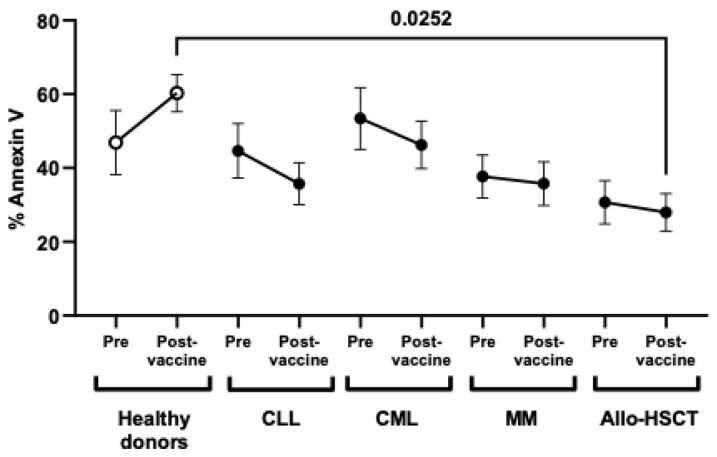
ADCC response against SARS-CoV-2 vaccine of PBMCs from individuals with OHD. Quantification by flow cytometry of the expression of phosphatidylserine by Annexin V in rituximab-coated Raji cells co-cultured with peripheral blood mononuclear cells (PBMCs) from individuals with chronic lymphocytic leukemia (CLL), chronic myeloid leukemia (CML), multiple myeloma (MM), or allogeneic stem cell transplant (Allo-HSCT), before and after receiving one dose of SARS-CoV-2 vaccine, in comparison with healthy donors. Each dot in the graphs corresponds to mean ± SEM. Statistical significance between groups was calculated using one-way ANOVA test.

**Figure 4 jcm-11-02803-f004:**
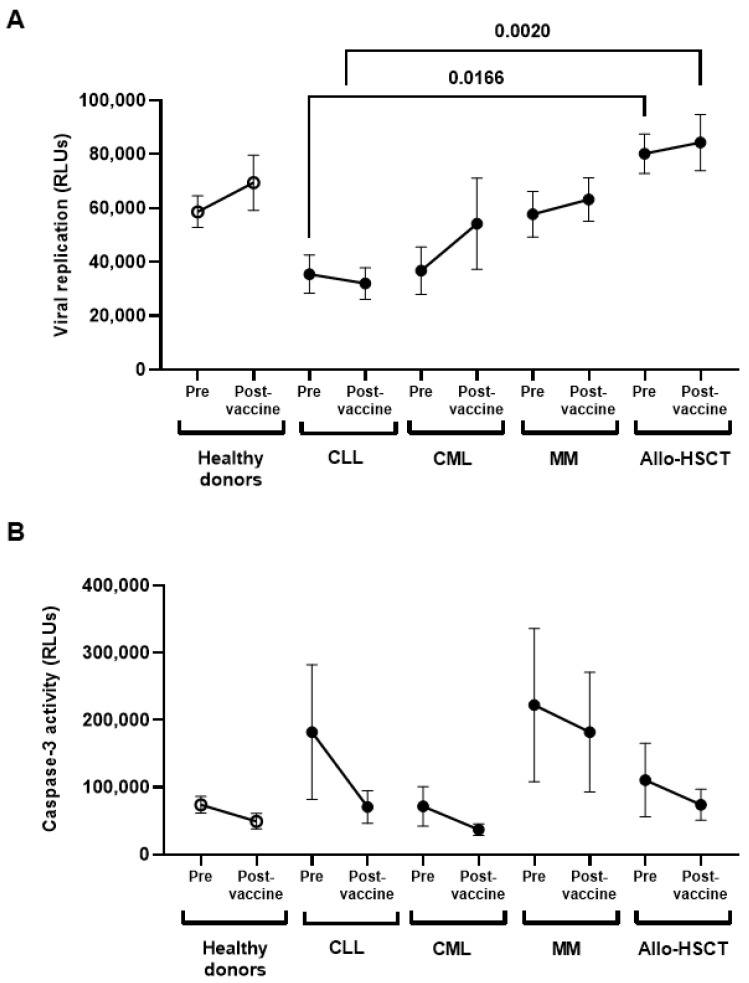
DCC and viral neutralization activity against SARS-CoV-2 infected cells of PBMCs of individuals with OHD before and after receiving one-dose vaccine. (**A**) The capacity of PBMCs from individuals with OHD to eliminate SARS-CoV-2-infected Vero E6 cells was determined by quantifying the production of Renilla (RLUs) after co-culture for 1 h. (**B**) DCC was assessed by measuring the activity of caspasa-3 in SARS-CoV-2-infected Vero E6 cells co-cultured with PBMCs from individuals with OHD, in comparison with healthy donors. Each dot in the graphs corresponds to mean ± SEM. Statistical significance between groups was calculated using one-way ANOVA test.

**Figure 5 jcm-11-02803-f005:**
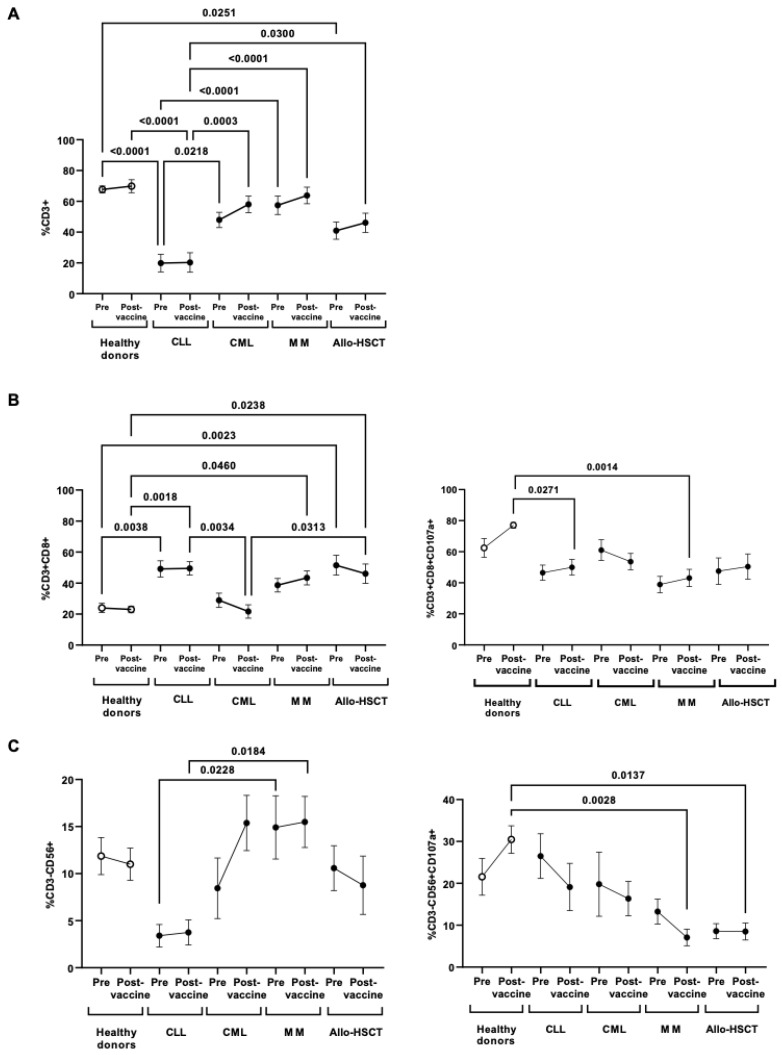
Characterization of the cellular cytotoxic populations in PBMCs from individuals with OHD before and after receiving one-dose vaccination. (**A**) Total CD3^+^ T cells in peripheral blood mononuclear cells (PBMCs) from individuals with oncohematological diseases (OHD) was evaluated by flow cytometry, in comparison with healthy donors. (**B**) CD8^+^ cell count (**left graph**) and expression of CD107a in these cells (**right graph**) was determined in PBMCs from individuals with OHD, in comparison with healthy donors. (**C**) NK cell count (**left graph**) and expression of CD107a in these cells (**right graph**) was determined in PBMCs from individuals with OHD, in comparison with healthy donors. Each dot in the graphs corresponds to mean ± SEM. Statistical significance between groups was calculated using one-way ANOVA test.

**Figure 6 jcm-11-02803-f006:**
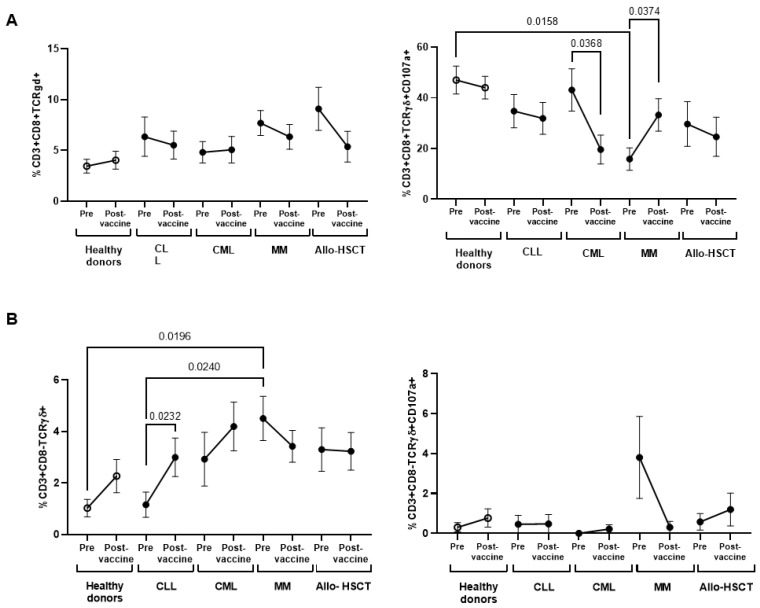
Characterization of CD3+CD8±TCRγδ+ cells in PBMCs from individuals with OHD before and after receiving one-dose vaccination. (**A**) CD3^+^CD8^+^TCRγδ^+^ cell count (**left graph**) and expression of CD107a in these cells (**right graph**) was determined in PBMCs from individuals with OHD, in comparison with healthy donors. (**B**) CD3^+^CD8^-^TCRγδ^+^ cell count (**left graph**) and expression of CD107a in these cells (**right graph**) was determined in PBMCs from individuals with OHD, in comparison with healthy donors. Each dot in the graphs corresponds to mean ± SEM. Statistical significance between groups was calculated using one-way ANOVA test and statistical significance within groups was calculated using Mann–Whitney test.

**Table 1 jcm-11-02803-t001:** Baseline demographic and clinical characteristics of the individuals with oncohematological diseases (OHD) recruited for this study.

	CLL (n = 15)	CML (n = 10)	MM (n = 15)	Allo-HSCT (n = 12)
Demographic characteristics				
Age, median (IQR)	65.9 (64.5–76.4)	61.9 (52–76.7)	72 (59.8–76.9)	60.1 (46.6–66-1)
Sex: male, n (%)	9 (60)	7 (70)	9 (60)	7 (58.3)
Days since first dose vaccination, median (IQR)	21 (18–33)	27 (21–28)	23 (18–27)	24 (20–28)
Treatment: n (%)	- BTK inhibitor: 4 (26.7) - Other treatments: 4 (26.7) - Watch and wait: 7 (46.7)	- TKI discontinuation: 4 (40) - Active treatment with TKI: 6 (60)	- Maintenance treatment after ASCT: 6 (40) - Non-ASCT candidates: 9 (60)	- Immunosuppresive treatment: 6 (50) - Non-immunosuppressive treatment: 6 (50)
Bone marrow transplant patients				
Type: n (%)	N/A	N/A	ASCT: 6 (40)	Allo-HSCT: 12 (100)
Months since transplant, median (IQR)	N/A	N/A	32 (23–60)	36 (21–44)
cGvHd, n (%)	N/A	N/A	N/A	9 (75)
Vaccines and analytics parameters				
Vaccine, n (%)				
Vaxzevria (AstraZeneca)	4 (26.7)	3 (30)	0	0
Spikevax (Moderna)	7 (46.7)	6 (60)	7 (46.7)	0
COMIRNATY (Pfizer)	4 (26.7)	1 (10)	8 (53.3)	12 (100)
Pre-vaccine, median (IQR) × 109/L				
Neutrophils	3.2 (1.6–4.8)	3.5 (3.1–4)	2.4 (1.4–2.9)	2.9 (2.2–4.4)
Lymphocytes	17.7 (1.1–79.8)	2.6 (1.9–2.9)	1.7 (1.2–2.4)	1.9 (1.3–2.9)
Monocytes	0.6 (0.3–0.8)	0.6 (0.4–0.8)	0.5 (0.4–0.7)	0.6 (0.4–0.8)
Platelets	119 (91.2–153)	241 (188.5–331.5)	160 (144–199.5)	197 (121.5–271.5)
Post-vaccine, median (IQR) × 109/L				
Neutrophils	4 (2–4.9)	4.6 (1.7–4.6)	2.3 (1.7–2.6)	3.2 (2.2–7.2)
Lymphocytes	18.9 (5.3–112.6)	1.8 (0.1–3.5)	1.7 (1–2.2)	1.7 (0.2–7.4)
Monocytes	0.9 (0.7–5.3)	0.5 (0.4–0.9)	0.5 (0.4–0.8)	0.6 (0.1–1)
Platelets	166 (122.7–255)	199 (134–361)	166.5 (122.2–203.5)	262.5 (195–477)

Allo-HSCT, allogeneic hematopoietic stem cell transplant; ASCT, autologous stem cell Transplant; BTK, bruton tyrosine kinase; cGvHd, chronic graft versus host disease; CLL, chronic lymphatic leukemia; CML, chronic myeloid leukemia; IQR, interquartile range; MM, multiple myeloma; N/A, not applicable; TKI, tyrosine kinase inhibitor.

## Data Availability

The original contributions presented in the study are included in the article. Further inquiries can be directed to the corresponding authors.

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
