# Peer review of "Early Cellular and Humoral Responses Developed in Oncohematological Patients after Vaccination with One Dose against COVID-19"

_jcm, 2022, doi:10.3390/jcm11102803_

Round 1

Reviewer 1 Report

This prospective study assessed the humoral and cellular response to first SARSCOV2 vaccine dose in 15 CLL (7 untreated 4 on BTKi), 10 CML (6 on TKI, 4 discontinued), 15 MM (6 on maintenance IMIDs) and 12 allotransplanted patients (6 not on IS) + 14 healthy donors and detected extraordinarily high humoral resposne in CML patients on TKI, while the poor humoral response in the other groups was highly compensated by enahanced ADCC functioning of T8 subgroups especially in CLL.

One major comment regards therapy-specific analyses. Untreated patients achieve better sercoversion (Borgogna 2022 Blood Cancer J), but the better serocoversion of CML patients is attributed to restored immune response in the many patients who discontinued TKI ("cured") or to those on TKI? Despite the small figures, a comparison is necessary. the opposite is expected fro those on BTKi, IMID or daratumumab.

Minor comments:

1) table 1: which treatment are MM patients non-ASCT candidates receiving?

2) table 1: please specify which treatments are 4 CLL patients receiving

3) figure 4: legend A) vs B) seem inverted

Reviewer 2 Report

The authors present early cellular and humoral immune response in patients with hematological cancers, after one COVID-19 vaccination. The authors present sound scientifics and data are presented in a clear way that make results easy to understand. The authors show that quantity is not equal to quality when it comes to IgG, as serocoversion does not always correlate with inhibitory activity and cellular immune response. Importantly, vaccination seems to be atleast partially usefull in the investigetad cohorts.

However, as I am sure the authors are aware of, there are some weaknesses. 

Since patients were included in February - June 2021 it should be possible to add clinical information on breakthrough infections, which would enhance clinical translation.

Around the world patients with hematological cancers are now offered 4th and 5th doses of COVID19 vaccination, which unfortunately makes data like these outdated. However, the authors bring interesting additional information, as compared to other publications.

The authors should also comment on a potential lack of protection against latest viral strains.

Also, numbers are small, as they often are when extensive laboratory analyses and work are performed.

Please explain all abbreviations when used for the first time. For example OHDs line 47, TKI line 52, MM line 62, ASCT line 65, HSCT line 68 and so on.
